# Characterization of In Vitro 3D Cell Model Developed from Human Hepatocellular Carcinoma (HepG2) Cell Line

**DOI:** 10.3390/cells9122557

**Published:** 2020-11-28

**Authors:** Martina Štampar, Barbara Breznik, Metka Filipič, Bojana Žegura

**Affiliations:** 1Department of Genetic Toxicology and Cancer Biology, National Institute of Biology, 1000 Ljubljana, Slovenia; martina.stampar@nib.si (M.Š.); barbara.breznik@nib.si (B.B.); metka.filipic@nib.si (M.F.); 2Jozef Stefan International Postgraduate School, 1000 Ljubljana, Slovenia

**Keywords:** 3D cell model, HepG2, cell proliferation, cell cycle, gene expression

## Abstract

In genetic toxicology, there is a trend against the increased use of in vivo models as highlighted by the 3R strategy, thus encouraging the development and implementation of alternative models. Two-dimensional (2D) hepatic cell models, which are generally used for studying the adverse effects of chemicals and consumer products, are prone to giving misleading results. On the other hand, newly developed hepatic three-dimensional (3D) cell models provide an attractive alternative, which, due to improved cell interactions and a higher level of liver-specific functions, including metabolic enzymes, reflect in vivo conditions more accurately. We developed an in vitro 3D cell model from the human hepatocellular carcinoma (HepG2) cell line. The spheroids were cultured under static conditions and characterised by monitoring their growth, morphology, and cell viability during the time of cultivation. A time-dependent suppression of cell division was observed. Cell cycle analysis showed time-dependent accumulation of cells in the G0/G1 phase. Moreover, time-dependent downregulation of proliferation markers was shown at the mRNA level. Genes encoding hepatic markers, metabolic phase I/II enzymes, were time-dependently deregulated compared to monolayers. New knowledge on the characteristics of the 3D cell model is of great importance for its further development and application in the safety assessment of chemicals, food products, and complex mixtures.

## 1. Introduction

In recent years, considerable efforts have been made to develop hepatic in vitro 3D cell models with higher predictability for detecting the genotoxic effects of chemicals and environmental particular liver, which is the main target organ of chemical activation and detoxification processes [1,2]. In toxicology, it is nowadays recommended to use alternative in vitro models for the implementation of the 3R (reduce, refine, and replace) strategy as well as considering the inaccurate prediction of animal models due to inter-species variability. This demands an urgent need for advanced, robust, cost, and time-efficient in vitro models for the safety assessment [3,4]. It is widely recognised that human primary hepatocytes (PHH) are the golden standard for studying metabolism and toxicity, as they are of human origin and express metabolic enzymes that are relevant for human metabolism of xenobiotics [5,6,7,8,9]. However, there are several shortcomings related to the application of PHH such as limited availability, the complexity of isolation and culturing of cells as their expansion in culture is not possible, short life span, rapid dedifferentiation, loss of many hepatocyte functions and hepatic phenotype when cultured in monolayer cultures, high costs associated with the performance of the experiments, and the most important—genetic and metabolic differences due to polymorphism of donors. Due to the above-mentioned facts, PHH are not suitable for routine use for genotoxicity testing [5,10,11,12]. Alternatively, hepatic carcinoma-derived cell lines, such as HepG2, C3A, HepaRG, HuH6, and many others, are frequently used in genotoxicity studies, due to their unlimited growth, availability, and high reproducibility of results. These cell lines [13,14,15,16] have several phenotypic characteristics and some functional properties of liver cells [17,18] and therefore, represent an effective compromise between the ease of culturing and the expression of several key enzymes involved in xenobiotic metabolism [19,20,21]. However, the major limitations of hepatic cells cultured in two (2D) dimensions are the low expression of CYP450 enzymes, xenobiotic receptors, and phase II enzymes [5,22] and thus, inadequate expression of liver cell function in vivo. Consequently, it is highly plausible that such cell models give inaccurate and false-positive results [23].

In drug development and hepatotoxicity research, there is a high demand for establishing new, reliable, and uniformed models [24] with higher predictability for the consequences of human exposure. The report from the Workshop on Genotoxicity Testing (IWGT) recommends focusing on the development of 3D models, which better reflect the in vivo conditions and are capable of creating the complex microenvironment with the purpose of advancing our understanding of complex biological phenomena [4]. Another important advantage of 3D cell models compared to 2D cell models is that they enable prolonged exposures, due to their increased stability as they retain high cell viability and morphology over the period of several weeks [25]. Many static and perfused techniques for culturing 3D cell models (spheroids) are available, such as non-adhesive surfaces, hanging drop cultures, spinner flasks, bioreactors, and micro-moulding [9,26,27]. Each of these techniques offers various advantages; however, many are technically challenging, expensive, and need appropriate facilities [28]. The most important advantages of culturing cells in the form of 3D are increased cell–cell and cell–matrix interactions and higher expression of liver-specific functions (albumin content, urea synthesis, and expression of cytochromes), thus providing a physiologically more relevant model in vivo [29,30,31,32,33,34,35]. In recent years, hepatic 3D cell models are, in addition to being used for drug development, also applied for studying genotoxic effects, such as chromosomal instability [28] and DNA damage [32,35,36] induced by various chemicals and environmental samples [37]. Although 3D cell models are superior to 2D cell models, the lack of standardisation of 3D cell culture protocols and insufficient characterisation of the spheroids prevent the integration of 3D cell culture models into the field of toxicology research, particularly due to variability in structure and composition of the formed spheroids [38,39].

The aim of the present study was to characterise the HepG2 cells grown in 3D conformation in terms of hepatic properties and the mRNA expression profile of selected genes coding for cell proliferation, drug-metabolising enzymes, transporters, and liver-specific factors. The HepG2 spheroids were formed by the forced floating method from an initial cell density of 6000 and 3000 cells/spheroid. As the age of the spheroids can influence the outcome and sensitivity of the cell model [40], spheroid growth, quality parameters (surface area and perimeter), and viability of cells were monitored over a period of 6 days and 12 days, respectively. The viability of cells in spheroids was determined by the Live/Dead staining using confocal microscopy, while the proliferation of cells (KI67 marker) in spheroids of different age and cell cycle analysis was assessed with flow cytometry. The mRNA expression of selected hepatic markers, genes involved in cell proliferation, and those involved in phase I/II metabolism in spheroids with an initial density of 3000 cells/spheroid was assessed using real-time quantitative PCR (Fluidigm).

## 2. Materials and Methods

### 2.1. Chemicals

Minimum essential medium eagle (MEME), penicillin/streptomycin, Na-pyruvate, non-essential amino acids (NEAA), l-glutamine, NaHCO_3_, dimethylsulphoxide (DMSO), methylcellulose, propidium iodide (PI), and fluorescein diacetate (FDA) were purchased from Sigma (St. Louis, MO, USA). Trypsin-EDTA (0.25%), foetal bovine serum (FBS), and TRIzol^®^ reagent were obtained from Gibco (Praisley, Scotland, UK). Hoechst 33258 dye was obtained from Invitrogen (Waltham, MA, USA). Phosphate-buffered saline (PBS), methanol, and ethanol were purchased from PAA Laboratories (Dartmouth, NH, USA). Triton X-100 was obtained from Fisher Sciences (Massachusetts, NJ, USA), while the high capacity cDNA archive kit, TaqMan Universal PCR Master Mix (4440038), and TaqMan Gene Expression Assays were from Applied Biosystems (Massachusetts, NJ, USA). The PreAmp GrandMasterMix (TA05-50) was obtained from TATAA Biocenter AB (Göteborg, Sweden). GE 48.48 Dynamic Array Sample & Assay loading Reagent Kit—10 IFCs (85000821) and 48.48 Dynamic Array: Gene expression chip were obtained from Fluidigm (South San Francisco). Anti-Ki-67-FITC (130-107-586) antibodies and REA Control (I)-FITC (131-104-611) were obtained from Miltenyi Biotec (Bergisch Gladbach, Germany).

### 2.2. Cell Culture and Formation of 3D Spheroids

The HepG2 cell line was obtained from the ATCC cell bank (HB-8065™) and was grown in MEME media supplemented with 10% FBS, 1% NEAA, 100 IU/mL pen/strep, 0.1 g/mL NaHCO3, 0.1 g/mL Na-pyruvate, and 2 mM l-glutamine at 37 °C in 5% CO_2_ atmosphere. The spheroids were developed by the forced floating method described in Štampar et al. (2019) [35] using a growth medium supplemented with 4% methylcellulose [41]. Two culture conditions for spheroids formation were used. The spheroids with an initial density of 3000 cells/spheroid and 6000 cells/spheroid were seeded and grown for 12 and 4 days, respectively, depending on the specific endpoint measured. The culture media was changed every 2–3 days to obtain the optimal growth of spheroids.

### 2.3. Monitoring the Growth and Morphology of Spheroids

The surface area of at least 10 spheroids with the initial density of 3000 cells/spheroid and 6000 cells/spheroid was monitored during the time of cultivation (up to 12 and 4 days, respectively). The surface area (mm^2^) of each spheroid was monitored microscopically every day and determined by planimetry at 40× magnification using the NIS elements software 4.13 v (Nikon Instruments, Melville, NY, USA) connected with the Ti Eclipse inverted microscope (Nikon, Japan). Evaluation of the spheroid growth was determined three times independently.

### 2.4. Quantification and Viability Determination of the Whole Spheroid by Live/Dead Staining

At least three spheroids for each condition (3000 and 6000 cells/spheroid) were stained and monitored during the time of cultivation. The culture media were replaced with FBS-free media supplemented with FDA (8 μg/mL) and incubated for 1 h in darkness at 37 °C in a 5% CO_2_ atmosphere. After staining, the cells were washed with PBS. Subsequently, PI (20 μg/mL) was added and incubated in the dark for an additional 5 min. Following the incubation, the staining mixture was washed with PBS and substituted with fresh serum-free MEME media (100 µL). The staining was performed according to [37] with modifications. The Z-stacks images of single spheroids were captured using the confocal laser scanning microscope Leica TCS SP8 at 100x magnification. Z-stacks of optical sections were captured across the entire spheroid thickness using excitation and emission (PI: 493/636 nm, FDA: 488/530 nm) settings for simultaneous dual-channel recordings; approximately 50 Z-stacks per spheroid were taken. Z-stacks were processed and analysed using the Leica confocal software and presented as a “maximum intensity projection image” gallery. Three independent experiments were performed (N = 3). The quantification of Z-stacks was proceeded in the program Image-Pro 10 (Media Cybernetics, Inc., Rockville, MD, USA), where at least 20 Z-stacks per spheroid were quantified. The percentage of dead cells in the spheroid was calculated as a ratio between the spheroid area and the number of dead cells.

### 2.5. Analyses of Cell Cycle and Cell Proliferation by Flow Cytometry

The analyses of cell cycle and cell proliferation in spheroids were performed with flow cytometry. The cells from spheroids with an initial density of 3000 and 6000/cells were collected during the time of cultivation for 18 and 4 days, respectively. For the analysis, 30 spheroids per sample were collected and dissociated with enzymatic digestion and mechanical degradation, as described by Štampar et al. (2019). The obtained single-cell suspension was washed (1x PBS), fixed in ethanol, and kept at −20 °C until the analysis (for details see Hercog et al. (2019 [42]; 2020 [37])). After the fixation step, the cells were washed in cold PBS, centrifuged, and labelled with anti-KI67-FITC (50-fold diluted antibodies in 1% BSA). Subsequently, the cells were washed with BSA and PBS and afterwards stained with Hoechst 33258 dye (diluted in 0.1% Triton X-100 1:500). The cell analyses were performed on a MACSQuant Analyzer 10 (Miltenyi Biotech, Bergisch Gladbach, Germany), where the FITC intensity, corresponding to the proliferation marker KI67+, was detected in the B1 (525/550 nm) channel and the analysis of the cell cycle was proceeded in the V1 (450/550 nm) channel. Rea-FITC control (Miltenyi Biotec, Bergisch Gladbach, Germany) was used to exclude unspecific binding. The experiments were repeated in three independent biological repetitions, where each time, 2.5 × 10^4^ single cells per experimental point were recorded. The obtained data were analysed and the graphics were prepared in FlowJo software V10 (Becton Dickinson, Franklin Lakes, NJ, USA). The cell cycle distribution of the solvent control and treated samples and KI67-positive cells for each day was compared to the first day of the measurements. The statistical analysis for cell cycle was performed by the two-way ANOVA with Bonferroni’s multiple comparisons test, alpha 0.05%, while the statistical significance of KI67-positive cells was assessed by one-way ANOVA with Dunnett’s multiple comparisons test, α0.05, both using GraphPad Prism V6 (GraphPad Software, San Diego, CA, USA).

### 2.6. The qPCR Analyses of the Expression of Selected Genes

For the gene expression analysis, only spheroids with a density of 3000 cells/spheroid were collected during the time of cultivation (every two to three days, starting with day three of cultivation). The basal expression of the selected genes was determined in HepG2 monolayer cultures (2D) cultured for 2 days (this is usually the time when in 2D, the gene expression is evaluated after the exposure to various compounds; 24 h for cells to attach to the surface of the plastic and 24 h of subsequent exposure) and in spheroids (3D) cultured for 3, 5, 7, 10, 12, 14, and 17 days. Total mRNA was isolated from one T25 plate in the case of 2D and from a pool of 25 spheroids per sample in the case of 3D, using the TRIzol reagent. The quality and the concentration of total mRNA were measured with a NanoDrop 1000 spectrophotometer (Thermo Fischer Scientific, Wilmington, DE, USA), while degradation was checked by gel-electrophoresis (BioRad Power PAC 3000 and UVP Chem Studio PLUS, Analytik Jena AG, Upland, CA, USA). The cDNA High Capacity Archive Kit was applied for the reverse transcription of 1 μg of total mRNA per sample. The expression of selected genes was quantified by the qPCR on 48.48 Dynamic Array™ IFC method, where TaqMan Universal PCR Master Mix and the pre-amplificated (TATAA PreAmp GrandMasterMix, Tataa Biocenter, Gothenburg, Sweden) Taqman Gene Expression Assays were used. To eliminate the effects of inhibition and to assess the performance of a primer set, a series of 5-fold dilutions of each target gene was analysed. The qPCR experiments were performed on 48.48 Dynamic Array™ IFC chips for gene expression on the BioMark HD machine system (Fluidigm, UK) and data were analysed with an open web program QuantGenious [43]. Experiments were repeated three times independently, each time in two replicates. The difference (2D versus 3D) in gene expression greater than 1.5-fold was considered as up/downregulation (relative expression >1.5 or <0.66, respectively).

The selected genes: *ALB* (albumin, Hs00910225_m1); *AFP* (α-fetoprotein), Hs00173490_m1; *ALDH3A1* (aldehyde dehydrogenase 3 family member A1), Hs00964880_m1; *TOP2A* (topoisomerase 2-α), Hs01032137_m1; *PCNA* (Proliferating cell nuclear antigen), Hs00427214_g1; *KI67* (cellular marker for proliferation), Hs01032443_m1; *CCND1* (encodes the cyclin D1 protein), Hs00765553_m1; *CDKN1A* (Cyclin Dependent Kinase Inhibitor 1A), Hs00355782_m1; *CYP1A2* (cytochrome P450 family 1 subfamily A member 2), Hs00167927_m1; *CYP1A1* (cytochrome P450 family 1 subfamily A member 1), Hs01054797_g1; *CYP3A4* (cytochrome P450 family 3 subfamily A member 4), Hs02514989_s1; *AHR* (aryl hydrocarbon receptor), Hs00169233_m1; *UGT2B7* (UDP glucuronosyltransferase family 2 member B7), Hs00426592_m1; *UGT1A1* (UDP glucuronosyltransferase 1 family, polypeptide A1), Hs02511055_s1; *SULT1C2* (sulfotransferase family 1C member 2), Hs00602560_m1; *SULT1B1* (sulfotransferase family 1B member 1), Hs00234899_m1; *NAT1* (N-acetyltransferase 1), Hs02511243_s1; *NAT2* (N-acetyltransferase 2), Hs01854954_s1; *HIF1α* (Hypoxia-inducible factor 1-alpha), Hs00153153_m1; *BBC3* (p53 upregulated modulator of apoptosis), Hs00248075_m1 were pre-amplificated. In all experiments, *GAPDH* (Human Endogenous Control, Hs99999905_m1) and *HPRT1* (Hypoxanthine phosphoribosyltransferase 1), Hs02800695_m1) were used as reference genes.

The different tests were performed between the 2 days old monolayer and spheroids on day 3 and all subsequent days with the two-way ANOVA (Dunnett’s test) by multiple unpaired t-test analysis using the Sidak–Bonferroni method (^+^
*p* < 0.05, * *p* < 0.05, respectively). The statistical significance within days was conducted by the two-way ANOVA, considering Tukey’s multiple comparisons test († *p* < 0.05).

## 3. Results and Discussion

The 2D cultures traditionally used for studying the genotoxic effects of chemicals have several limitations, which consequently lead to misleading results. This has become even more evident with the development of in vitro models that enable the growth of cells in three dimensions (3D), which is physiologically more similar to in vivo conditions [44]. However, before 3D cell models can be integrated for genotoxicity testing research, there is a need for the development and subsequent standardisation of robust models that accurately predict the possible effects of studied compounds [45].

To our knowledge, the present study is the first where a comprehensive characterisation of the HepG2 3D cell model was performed. Based on previous results, two initial cell densities of 3000 and 6000 cells per spheroid were selected for the development of 3D cell models [35]. The HepG2 spheroids were formed by the forced floating method and cultured under static conditions for several days. During the time of cultivation, the spheroids were characterised by measuring the surface area, and viability of cells in spheroids. The characterisation is a very important step in the development of 3D cell models as the obtained data provide the information on whether the model is comparable to the liver in vivo conditions and is thus, more accurate than traditional monolayer culture.

### 3.1. The Effects on Growth and Morphological Changes over Time

The growth of at least 10 spheroids with the initial density of 3000 cells/spheroid (Appendix A) and 6000 cells/spheroid (Appendix A) was daily measured from 24 h to 7 days. The surface area was monitored by light microscopy and planimetry. The size of the spheroids varied according to the number of cells seeded in each well and increased with time of incubation. The spheroids with an initial density of 3000 cells/spheroid (Appendix A) grew time-dependently from the first day of seeding. At the end of the cultivation (7 days), the surface area (0.39 ± 0.01 mm^2^) increased 95% compared to the surface area at 24 h (0.2 ± 0.02 mm^2^) (Appendix A). In contrast to this, the spheroids with an initial density of 6000 cells/spheroid (Appendix A) grew slower and the surface area after 7 days of cultivation increased for only 20% (0.56 ± 0.03 mm^2^) compared to 24 h (0.45 ± 0.05 mm^2^) (Appendix A). At both initial densities, the spheroids maintained uniform spherical shape over the course of the culturing. These results show a steady growth of spheroids during the time of culture, which is in line with other studies on HepG2 spheroids [18,32]. The increase in the average surface area was higher at the lower initial cell density, meaning that the cells proliferated at a higher rate compared to spheroids with higher initial density, which was also reported by Lee et al. (2009) [46]. Based on these results, we concluded that 3000 cells per well was the more optimal density for the formation of spheroids that could be used for long-term exposures; therefore, this density was selected for further characterisation of the spheroids.

### 3.2. Determination of Live/Dead Cells in Spheroids over Cultivation Time

Live/Dead staining enabled us to investigate at which initial density and time of cultivation the viability of cells started to decrease and consequently, form the necrotic core. A very well-known limitation of aged spheroids is the formation of a necrotic core, which results from the accumulation of metabolic waste products and insufficient diffusion of oxygen/nutrients into the centre of the spheroid starting at a diameter above 200 to 500 µm, as reported by Nath and Devi (2016) [47]. The visual analysis of HepG2 spheroids by confocal microscopy verified the time-resolved viability of the cells in a 3D culture, which was more or less stable over the cultivation time. In Figure 1A,B, representative spheroids from day 3 until day 12 and day 6 in the case of 3000 and 6000 cells/spheroid, respectively, stained with FDA (live) and PI (dead) are shown. Although the percentage of PI-positive cells in spheroids with an initial density of 3000 cells/spheroid increased from 10 days of cultivation onwards, we noticed that only a few dead cells were observed in the centre of the spheroids, meaning that no necrotic core was formed. The quantification of PI-positive cells representing dead cells confirmed a time-dependent increase in non-viable cells that was significant after 10 and 12 days, reaching 14.5% and 18.9% on average (Figure 1C,D), respectively. In spheroids with an initial density of 6000 cells/spheroid, a higher percentage of dead cells compared to 3000 cells/spheroid was observed already after 3 days of cultivation that was 8.4% and 4.0% on average, respectively. Correlating these results with the spheroid growth data, the spheroids with an initial density of 3000 cells/spheroid gradually increased in size and stayed uniformly spherical with limited degrees of necrosis up to day 12. However, in larger spheroids, small patches of death cells started to occur approximately on day 4 and by day 6, reached 16.5% dead cells on average with the visible formation of a necrotic core. Spheroids usually consist of three main zones—an outer proliferating rim, a quiescent viable zone, and an inner necrotic core that can develop due to lower diffusion of nutrients and oxygen forming hypoxic conditions [48,49,50]. In cancer research, the zones in larger tumour spheroids with a necrotic core resemble the cellular heterogeneity of solid in vivo tumours [51] and the necrosis occurring in the centre of the spheroid is a desirable characteristic as it mimics in vivo conditions. However, in genetic toxicology, where a model has to recapitulate an in vivo-like liver microenvironment, this is an undesirable characteristic. There are only a few studies so far that specifically determined the size or time at which hepatic spheroids develop necrosis that is associated with the cell type, cell number, and culture conditions [32,52]; therefore, the present study contributes new knowledge on the formation of a necrotic core in spheroids with time of cultivation. Previously, in encapsulated 3D HepG2 aggregates, no necrotic core was observed up to three days of culturing, while with prolonged cultivation, the thickness of aggregates disabled the determination of necrotic cells in the centre of the 3D cell model [19]. Elje et al. reported [32] a small necrotic core in HepG2 3D spheroids developed in hanging drops after 1 day of culturing, which was more represented as separate dead cells; however, the viability was stable over time and the cells were cultured for up to 21 days. Furthermore, in hepatic C3A spheroids with initial density of 2500 cells/spheroid, small patches of cell death started to occur approximately at day 14 and, by day 18, a necrotic core was formed [52]. Altogether, these studies, including ours, show that necrosis occurs in the core of the spheroid with time of cultivation and depends on the cell type and 3D conformation.

### 3.3. Distribution of Cells within the Cell Cycle and Cell Proliferation during the Time of Cultivation

Previous studies on HepG2 3D models reported a strong decrease in cell proliferation, which is associated with a time-dependent cell differentiation process [32,53]. In our study, the proliferation of cells and their distribution within the cell cycle of spheroids with the initial density of 3000 and 6000 cells/spheroid were determined by flow cytometry over the time of cultivation, with simultaneous detection of the fluorescent signals of FITC corresponding to the proliferation marker KI67 and Hoechst 33258 corresponding to the cell cycle distribution. It is well known that KI67 protein is present during all active phases of the cell cycle (G1, S, G2, and M), and it is absent from the resting cells (G0) [54,55]. Therefore, the novel approach of simultaneous staining allowed us to further distinguish the distribution of cells in the G0 (non-proliferating) and G1 (proliferating) phase. The results showed that the overall number of proliferating cells and the number of proliferating cells within the G0/G1 phase decreased over the time of cultivation, which is clearly the consequence of the time-dependent increase in non-proliferating cells (G0) within the G0/G1 peak (Figure 2A–C). The percentage of proliferating cells gradually declined with the time reaching 50% decrease after approximately 7 and 2 days of cultivation at an initial density of 3000 (Figure 2A) and 6000 (Figure 2B) cells/spheroid, respectively. At higher cell density, we noticed that already after 24 h, only 65.5% of cells proliferated and by day 4, only 28% of cells expressed KI67 protein, meaning that at day 4, the majority of cells did not proliferate. On the contrary, at lower initial cell density, the decrease in cell proliferation was slower. After 3 days of spheroid cultivation, 82% of cells proliferated, while with further cultivation, the proliferation rate decreased to approximately 68%, 54%, and 13% after 5, 7, and 18 days, respectively. The same trend was observed for proliferating cells within the G0/G1 phase. Similarly, a decrease in the number of proliferating cells in HepG2 spheroids and spheroids cultured in hydrogels was reported by Ramaiahgari et al. (2014) [31] and Lee et al. (2009) [46]. Our results clearly showed that spheroids with lower initial density maintain the proliferation of cells at a higher rate and over the longer period compared to the spheroids with higher initial density, meaning that spheroids with lower initial density are more suitable for long-term exposures.

The cell cycle duration determines the unique doubling time of the cells [26] that is coordinately controlled by cyclin-dependent kinases and their cyclin partners, whose levels fluctuate throughout the phases of the cell cycle [56] (Figure 2C). In the G1 phase of the cell cycle, crucial decisions on DNA replication and completion of the cell division are made [57]. From the obtained results, we can see that at the initial density of 3000 cells/spheroid, the distribution of cells by phases does not change over time of cultivation. At all measured time points during the time of cultivation, approximately 67.7 ± 3.4% of proliferating cells were in the G1 phase, 14 ± 1.8% of cells were in the S phase, and 15.8 ± 1.7% were in the G2 phase, meaning that the cell cycle was not disturbed during 18 days, only the ratio of proliferating/non-proliferating cells in G0/G1 cells changed. On the contrary, in spheroids with the initial density of 6000 cells/spheroid, a slight increase in proliferating cells in the G1 phase over the time of cultivation was observed, with 61.3% of cells in G1 after 24 h reaching up to 73.9% after 4 days, with a concurrent decrease in cells in the S and G2 phase from 21% and 17% after 24 h to 13.8% and 11.77% after 4 days, respectively (Figure 2D), suggesting cell cycle arrest and thus, the accumulation of cells in the G0/G1 phase with a time of spheroid cultivation. The cell cycle arrest can lead to the inhibition of cell proliferation and/or apoptosis [56], which is in line with the obtained data of the KI67 proliferation marker. These results again show that spheroids with lower initial density maintain a higher rate of cell proliferation compared to spheroids with higher initial density.

### 3.4. Gene Expression in Spheroids

For the first time, analysis of the expression of genes involved in the proliferation (*TOP2A*, *PCNA*, *KI67*, *CCND1*, and *CDKN1A*), apoptosis (*BBC3*), genes of hepatic markers (*AFP* and *ALB*), and gene a transcription factor encoding aryl hydrocarbon receptor (*AHR*), phase I (*CYP1A1*, *CYP1A2*, and *CYP3A4*), and phase II (*UGT1A1*, *UGT2B7*, *SULT1B1*, *SULT1C2*, *NAT1*, and *NAT2*) xenobiotic-metabolising enzymes was studied in HepG2 spheroids. Gene expression was evaluated in spheroids formed from the lower initial density, namely 3000 cells/spheroid that were cultured for 3, 5, 7, 10, 12, 14, and 17 days. Data for the individual genes were compared to the expression of genes from the monolayer culture at the age of 2 days and are presented as the ratio between 2D (at 2 days of culturing) and 3D at the corresponding time point.

#### 3.4.1. Expression of Genes Involved in Cell Proliferation

Cell proliferation can be regulated by several factors, such as mitogens, growth factors, and survival factors [58]. We studied the genes *PCNA, KI67*, and *TOP2A* over the time of culturing, as the expression of the genes involved in the process of cell proliferation may be affected by the time the cells are in culture. A proliferating cell nuclear antigen (*PCNA*), a protein with an important role in DNA replication, has many cell cycle-dependent properties and its absence leads to cell cycle arrest in the S and G2/M phase [59]. Another important proliferation marker KI67 encodes a nuclear antigen during the G1, S, and G2–M phases of proliferating cells, meaning that it is present during all active phases of the cell cycle, except the G0 phase [60,61]. The expression of *TOP2α* is cell cycle-dependent and encodes DNA topoisomerase, which is an enzyme that controls and alters the topologic states of DNA during transcription [60,62]. The results showed that all three studied genes, *TOP2A*, *PCNA*, and *KI67*, were time-dependently downregulated compared to 2 days old monolayer culture (Figure 3A). The major shift in the downregulation of the proliferation markers occurred approximately at day seven.

The changes in *KI67* on the mRNA level showed the same trend as the data obtained with flow cytometry, where a decrease in KI67-positive cells over the time of cultivation, with approximately 50% of proliferating cells, was detected at day seven. As described before, HepG2 cells accumulated in the G1/G0 phase, due to arrested division in the G0 phase, which is especially noticeable at the initial density of 6000 cells/spheroid. The low proliferation rate in the 3D model can therefore be effectively utilised for studying the effects of long term exposure to various compounds, which is not feasible in a 2D model as the confluence limits the duration of culturing [31,63]. A decrease in cell proliferation in different 3D cell models overtime was reported in several studies with HepG2, using extra-cellular matrix-based hydrogel [31] and spheroids prepared by the hanging drop method [32], and HepG2/C3A, culturing in rotating bioreactors [30]. The occurrence of reduced proliferation over the time of cultivation may also have an impact on the decreased expression of cyclin protein coding genes that control cell cycle progression [64]. In our study, the expression of cyclin protein coding gene *CCND1* was downregulated and a decrease in the first seven days was noticed from a −4.76-fold change after 3 days of cultivation to −8.33-fold change after 7 days of cultivation (Figure 3B). The *CCND1* encodes the cyclin D1 protein, whose elevated expression has been reported in many human tumours and correlates with increased cell proliferation and differentiation due to the reduced G1/S transition [65]. The expression of *CDKN1A*, which is the cell cycle-related gene responsible for the G2/M checkpoint [66], was downregulated (−9.09-fold at day 3) compared to monolayer culture. We noticed a slight increase in *CDKN1A* expression with time of incubation reaching the maximal level at day 7 (−7.14-fold) (Figure 3B). After the seventh day, the expression of both cyclin protein coding genesremained unchanged, which, compared to 2D monolayer culture, indicates the non-proliferating differentiated phenotype of HepG2 spheroids as described by Hiemstra et al. (2019) [67] and by Ramaiahgari et al. (2014) [31].

Altogether, the results suggest that the expression of gene markers related to the proliferation and division of cells decreased over time. Similarly, the expression of *BBC3*, a gene related to apoptosis, was downregulated in spheroids at the age of 3 days (−8.7-fold) compared to monolayer culture; however, with further incubation, the expression of *BBC3* increased, reaching the highest level after day 12 (−2.56-fold change) and then, declining again (Figure 3C). The pro-apoptotic protein BBC3 interacts with anti-apoptotic Bcl-2 family members, resulting in mitochondria-induced apoptosis and cell death through the caspase cascade [68]. It was reported that the diffusion of oxygen into the centre of spheroids is difficult if a diameter is larger than 200 µm (larger spheroids), which consequently causes hypoxia in the core [49]. Live/Dead stained spheroids (density 3000 cells/spheroid) showed no signs of necrosis in the centre; however, in spheroids with a higher initial density (6000 cells/spheroid), necrosis was observed at day 6. This could be related to the hypoxia as reported by Ramaiahgari et al. (2014) [31] (Figure 3). We further investigated the transcription level of the *HIF1α* gene, one of the hypoxia-inducible factors, which mediates cellular adaptation to hypoxic conditions [69]. The obtained results showed that at lower initial density (3000 cells/spheroid) during 17-day cultivation, the *HIF1α* was downregulated compared to monolayer culture and the changes were significantly different from day 7 onwards with no further time-dependent deregulations (Figure 3C), which indicates that no excessive hypoxia was present in HepG2 spheroids up to 17 days.

#### 3.4.2. Expression of Genes Encoding Hepatic Markers

Long-term cultivation of HepG2 cells in 3D conformation led to the enhancement of the expression of liver-specific markers, as shown for alpha-fetoprotein (*AFP*) and albumin (*ALB*) compared to monolayer culture (Figure 4). Alpha-fetoprotein is involved in pleiotropic activities affecting the processes of cell differentiation and growth regulation [70]. Compared to 2D, *AFP* expression in spheroids was 4.57-fold higher, which was statistically significant, at the age of 3 days and reached the maximal mRNA level at day 7 (6.84-fold). With further cultivation, the expression of *AFP* did not change much. Similarly, the expression of *ALB* in 3D significantly differed from the expression in 2D and was 4.95-fold upregulated at day 7. The *ALB* expression gradually increased and reached the maximal level at day 10 (9.56-fold) with a slightly decrease after 12 and 17 days (7.67-fold and 8.03-fold, respectively). Albumin is a stable protein and has a serum half-life of 20 days. Its synthesis is typically regulated on the transcriptional level [71]. Previously, the elevated expression of albumin in HepG2 spheroids compared to monolayer cultures was reported at mRNA [22,35] and protein [31] levels. Altogether, these studies show that hepatic functions are strongly enhanced in 3D systems compared to the 2D monolayer cultures [15,18,19,25,38,53].

#### 3.4.3. Expression of Genes Involved in the Xenobiotic Metabolism

Biotransformation of xenobiotic substances is divided into phase I and II reactions, where phase I reactions include the transformation of a parent compound to more polar metabolite(s), while phase II biotransformation results in metabolic inactivation by conjugating reactions including glucuronidation, sulfation, acetylation, methylation, glutathione, and amino acid conjugation. In general, the respective conjugates are more hydrophilic than the parent compounds and are excreted from the organism [72,73]. The most important enzymes from phase I reactions belong to the superfamily CYP450 and play an important role in cellular metabolism and homeostasis, and detoxification and metabolic activation of xenobiotic compounds into reactive metabolites [74,75]. In general, hepatocellular carcinoma monolayer cell cultures exhibit low expression of drug-metabolising enzymes including CYPs and therefore, do not represent ideal alternative systems to human hepatocytes for drug metabolism and hepatotoxicity testing [76]. Recently, several published studies have described the higher expression of CYP enzymes in hepatic cells cultured in 3D conformation [6,18,21,25,28,31,35,77]. Moreover, it has been reported that HepG2 spheroids developed by the forced floating method are metabolically competent, thus expressing various CYP450 enzymes, and are sensitive for detecting the genotoxic effects of indirect-acting genotoxic compounds [35]. In the present study, we evaluated the gene expression of *CYP1A1*, *CYP1A2*, *CYP3A4*, *AHR*, and *ALDH3A1* in 3D spheroids at the age of 3, 5, 7, 10, 12, 14, and 17 days, and compared to the expression of monolayer culture at the age of 2 days (Figure 5). The results revealed that all studied CYPs were clearly upregulated in a 3D model compared to 2D (Figure 5A). The basal mRNA level of *CYP1A1* was approximately 2-fold higher than in monolayer culture and remained at the same level throughout the whole duration of the spheroid cultivation. Similarly, the expression of *CYP1A2* remained constant from day 5 onwards. *CYP3A4*, the major hepatic CYP contributing to the metabolism of more than 50% of xenobiotic substances [76], was reported not to be expressed in HepG2 cells grown in monolayer [20], which strongly limits the use of HepG2 cells for the assessment of the drug metabolism. As already previously shown [35,67,78], the results of the present study confirmed the significant upregulation of *CYP3A4* from day 3 onwards (2.16-fold) with a gradual increase in the mRNA level, reaching the highest level at day 12 (5.76-fold). Altogether, these results demonstrate that HepG2 cells grown in 3D conformation show differentiation into more metabolically competent cells when compared to monolayer cultures. The expression pattern of metabolic genes closer to primary hepatocytes has also been reported for several hepatocellular carcinoma cell lines grown under certain prolonged culturing conditions. For instance, the confluent growth of Huh7 cells resulted in a cell phenotype change and an increase in the *CYP3A4* mRNA level, protein content, and activity after 4 weeks in confluent culture [79], while differentiated HepaRG cells expressed higher levels of CYPs compared to non-differentiated [80,81]. Additionally, HepG2 cells grown in 3D conformation expressed an increased level of *CYP1A1*, *CYP1A2*, and *CYP3A4* genes [35], while HepG2/C3A cells grown in 3D under dynamic conditions for 3 weeks showed elevated expression of *CYP1A1* and *CYP3A4* [77] compared to 2D cultures.

Another phase I enzyme ALDH3A1 (Figure 5C), a member of aldehyde dehydrogenase 3 family that catalyses the aliphatic and aromatic aldehydes to the corresponding acids [82,83], was downregulated already in 3 days old spheroids (−3.45-fold) and with further cultivation, the expression of *ALDH3A1* dropped sharply (−12.8-fold and −20-fold at 10 and 14 days, respectively). Low basal expression of *ALDH3A1* in the normal liver was reported by Muzio et al. (2012) [84], while induced levels were determined in liver, colon, bladder, and lung [82,85,86]. The results showing low gene expression are in line with the decreased proliferation of HepG2 cells in spheroids, as it is known that the activation of ALDH3A1 stimulates the proliferation [84], thus the low mRNA level of *ALDH3A1* corroborates decreased cell proliferation and arrested cell cycle determined by the flow cytometry. The expression of drug-metabolising enzymes including *CYP1A1, CYP1A2,* and *ALDH3A1* is regulated by the activation of nuclear factors such as aryl hydrocarbon receptors (AhR) [87]. The basal expression of the *AHR* gene in HepG2 spheroids at the age of 3 days did not significantly differ from the expression in monolayer culture; however, with further incubation, it slowly increased over the time of incubation, reaching the maximal level (1.41-fold) at 17 days. Altogether, the obtained results are in line with the findings of Ramaiahgari et al. (2014) [31], Shah et al. (2018) [28], and Štampar et al. (2019 [35]; 2020 [77]), who described that HepG2 spheroids express higher mRNA levels of phase I metabolic enzymes, which is an important physiological function of hepatic cells in vivo [88].

The enzymes from phase II drug metabolism represent the detoxification step of xenobiotic compounds with the main pathway by the formation of glucuronide conjugates by glucuronosyltransferases (UGTs). In HepG2 spheroids (Figure 5B), the basal mRNA level of *UGT1A1* compared to monolayer culture showed great upregulation from day 3 onwards that was time-dependent, reaching a maximal level at day 12 (5.74-fold). Subsequently, the *UGT1A1* mRNA level decreased (4.13-fold at day 14) and remained constant during further cultivation. Time-dependent elevation of *UGT1A1* expression in HepG2 spheroids was already previously reported by Ramaiahgari et al. (2014) [31] and Štampar et al. (2019) [35]. For the gene *UGT2B7*, slight but not significant upregulation was detected at day 3 (1.49-fold); however, its expression did not increase with time. Previously, it has been demonstrated that dynamic HepG2/C3A spheroids expressed an increased mRNA level of *UGT2B7* after three weeks of culturing [77]. In addition to glucuronidation, sulfonation is very important in the biotransformation of xenobiotics [89], where sulfotransferases (SULTs) catalyse the sulfate conjugation of a variety of exogenous chemicals and endobiotics using 3′-phosphoadenosine-5′-phosphosulfate as the donor [90]. On the other hand, in the reaction, pro-carcinogens are converted into highly reactive intermediates that can act as chemical mutagens and carcinogens by covalently binding to DNA [91]. The results showed that the mRNA level of *SULT1B1* was significantly upregulated (2.89-fold) already after 3 days of culturing; however, the expression was downregulated after prolonged cultivation (−2.13 fold and −3.33-fold at 12 and 17 days, respectively). On the contrary, *SULT1C2* was significantly elevated already at day 3 (3.09-fold) and the expression sharply increased with time (6.16-fold by day 17). Another important group of phase II enzymes is *N*-acetyltransferases (NAT) that catalyse the activation of aromatic and heterocyclic amines via *O*-acetylation, while *N*-acetylation of the parent amines is considered a detoxification step [73]. The gene encoding *NAT1* was not importantly deregulated in HepG2 spheroids—at most, it was slightly downregulated (−1.5-fold at day 17). In contrast, the expression of *NAT2* was significantly elevated with the highest expression observed at day 12 (3.95-fold). Transcriptomic analyses revealed that in dynamic HepG2/C3A spheroids, *NAT1* was not importantly deregulated when compared to 2D culture, while *NAT2* was significantly downregulated in 25 days old culture [77]. NATs are substrate-specific and have distinct tissue distribution, where NAT1 has a ubiquitous tissue distribution and its expression is related to cancers, while NAT2 activity has been described in the liver, colon, and intestinal epithelium [73]. In the present study, many crucial genes that are involved in the activation and detoxification of xenobiotic substances and are in HepG2 monolayer cultures expressed at a very low rate, or are even not detectable, were clearly expressed in HepG2 3D spheroids.

In a study on HepG2 spheroids, Elenberger et al. (2018), based on ultrastructural and organo-typic functional investigations, identified clearly different phases of HepG2 spheroids, including an early phase (day 3 to 6), mid-stage phase (day 6 to 12), and late phase (day 15 to 18), all showing significant differences in cell-to-cell interactions, specialised microstructures such as the formation of bile canaliculi, and metabolic activities including albumin and urea secretion. Similarly, the results of the present study suggested three stages of spheroid formation; the early stage at the age of 3–6 days, mid-stage at the age of 7–12 days, and the late stage at the age of >14 days. In HepG2 spheroids, the expression of hepatic markers and metabolic genes from phases I and II changed over time of cultivation with important changes observed by day 7. The highest mRNA levels for the majority of metabolic genes were noticeable at 10 to 12 days, which, with further cultivation, gradually declined, suggesting that the aforementioned cultivation time is long enough for HepG2 cell differentiation and the development of a metabolically competent cell model with the quantitative and qualitative expression of phase I and II metabolic enzymes compared to a 2D cell model. In addition to metabolic activity, we noticed an important change in cell proliferation at day 7 with a slight increase in cell death, which turned out to be significant after 10 days of cultivation. Taken together, our study demonstrated that the spheroidal age needs to be considered as an important parameter in the development of spheroid-based in vitro models.

## 4. Conclusions

In vitro, 3D cell models compared to traditional monolayer cultures better resemble the cell organisation of tissues and organs and thus, more accurately mimic the in vivo microenvironment. In recent years, they have also become a promising tool in the field of genetic toxicology in order to reduce, replace, and refine animal experiments. Nevertheless, before 3D models can be routinely used for genotoxicity assessment, they have to be comprehensively characterised and growth conditions need to be optimised to allow for the reproducibility and comparability of the results. Furthermore, systematic characterisation allows us to identify all the crucial advantages and disadvantages, which is of high importance for the further use of the new 3D models. To our knowledge, the present study is the first where a 3D HepG2 cell model was systematically characterised and standardised including advanced cell cycle and proliferation analysis by flow cytometry, and gene expressions. In the present work, we developed uniform 3D HepG2 spheroids of similar size and shape grown under static conditions. The influence of spheroidal age on cell proliferation and metabolic status was studied over a 17-day cultivation period to gain a deeper understanding of the morphological and physiological characteristics of HepG2 spheroids. Based on new knowledge obtained within our study, we can conclude that the initial cell density for the formation of spheroids is very important in order to obtain spheroids with viable dividing cells, which is a prerequisite for studying the adverse geno-/toxic effects, as a division of cells is necessary for damage to be incorporated into DNA. Compared to the 2D monolayer cultures, HepG2 spheroids showed a time-dependent reduction in cell proliferation with cell division arrested in the G0/G1 phase of the cell cycle. Moreover, the spheroids revealed increased liver-specific functions and demonstrated strong physiological relevance concerning gene expression of hepatic markers and metabolic enzymes, in particular for sulfotransferases in phase II, thus indicating differentiation into more metabolically competent cells; this, however, has to be further confirmed at the protein level. We believe that the 3D HepG2 cell model with characterised cell growth and proliferation, as well as known expression of hepatic markers and metabolic enzymes, will contribute to a more reliable assessment of genotoxic activity of chemicals, due to its higher physiological relevance for human exposure and may, therefore, provide an alternative to animal models, which comply with the 3Rs policy to reduce in vivo testing.

## Figures and Tables

**Figure 1 cells-09-02557-f001:**
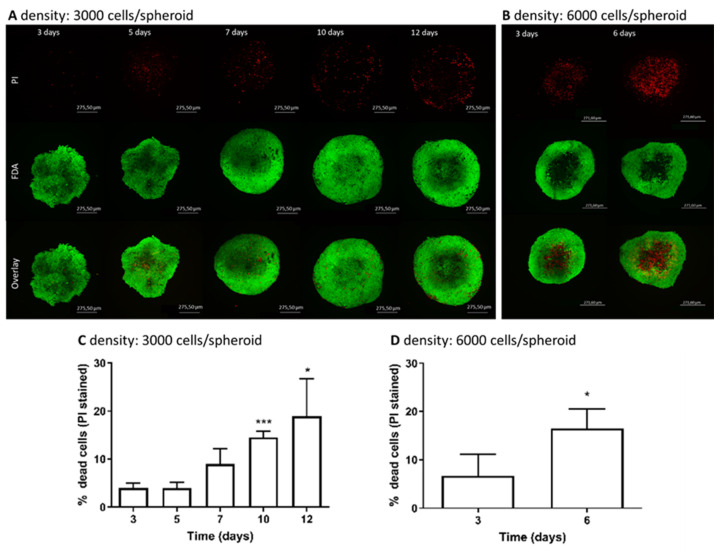
Images of Live/Dead-stained spheroids(**A**,**B**) and the quantification of the percentage of PI-positive cells in images of spheroids (**C**,**D**) captured over the period of cultivation at different densities (**A**–**C**) 3000 cells/spheroid and (**B**–**D**) 6000 cells/spheroid. The cells in spheroids were stained with FDA (green, live cells) and PI (red, dead cells). The Z-stacks were obtained using a confocal microscope at 100× magnification (N = 3). A “maximum intensity projection image” of the spheroid was generated from 50 Z-stacks images (**A**,**B**). Z-stacks were quantified with the Image-Pro 10 software (**C**,**D**) where at least 20 Z-stacks per spheroid were measured and the percentage of dead cells in the spheroid was calculated (N = 3). The statistical significance was calculated with the Student t-test, with alpha 0.05, * *p* < 0.05, ** *p* < 0.01, *** *p* < 0.001, **** *p* < 0.0001.

**Figure 2 cells-09-02557-f002:**
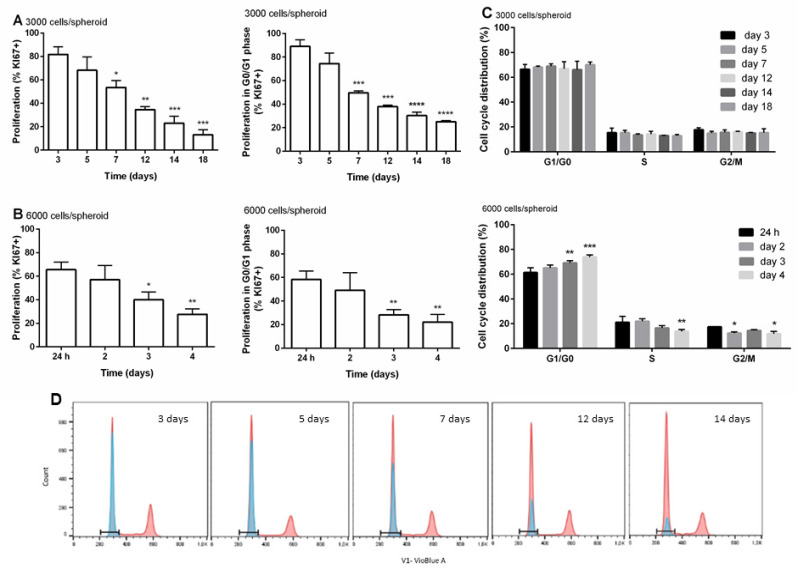
The proliferation of cells and the distribution of cells in different phases of the cell cycle during the time of cultivation. Percentage of KI67-positive cells and the percentage of KI67-positive cells within the G0/G1 phase of cell cycle overtime at density 6000 cells/spheroid (**A**). Percentage of KI67-positive cells and the percentage of KI67-positive cells within the G0/G1 phase of cell cycle overtime at density 3000 cells/spheroid (**B**). Distribution of cells in different phases of the cell cycle over the period of cultivation (**C**). Representative overlays of simultaneous staining with Hoechst for the cell cycle (red) and anti-KI67 antibody (blue) (**D**). The results are presented as the mean ± SD (N = 3). The statistical analysis was conducted in GraphPad Prism 6, by the two-way ANOVA using the Bonferroni multiple comparisons test, * *p* < 0.05, ** *p* < 0.01, *** *p* < 0.001, **** *p* < 0.0001.

**Figure 3 cells-09-02557-f003:**
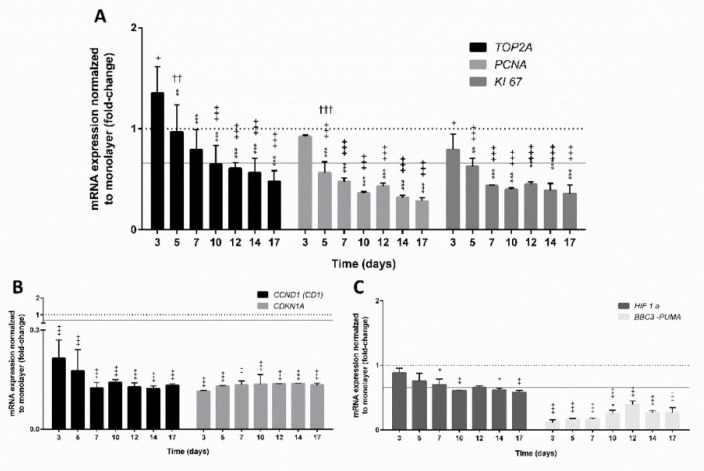
Relative expression of genes involved in the proliferation process (**A**,**B**) and apoptosis (**C**) over time from 3 to 17 days. The significant difference between (i) the monolayer culture (2D) and the spheroids (3D) ( + *p* < 0.05, ++ *p* < 0.01, +++ *p* < 0.001); (ii) the first day (at day 3) of measurement and all subsequent days (* *p* < 0.05, ** *p* < 0.01, *** *p* < 0.001); and (iii) within days (†† *p* < 0.01, ††† *p* < 0.001) was calculated in GraphPad Prism 6, by two-way ANOVA considering Dunnett’s multiple comparisons test. Results are presented as the mean ± SD (N = 3). The dotted line denotes the expression of the corresponding gene in monolayer culture (1-fold change), the grey line indicates up- or downregulation of genes with the threshold set at 1.5-fold, which is more than 1.5 or less than 0.66 relative expression, respectively.

**Figure 4 cells-09-02557-f004:**
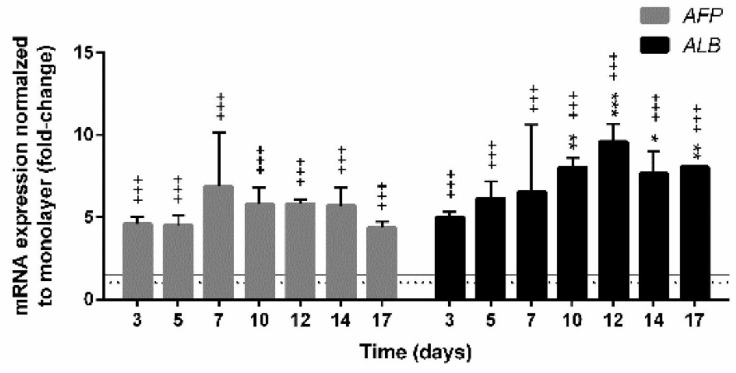
Monitoring the mRNA expression of hepatic markers over time. The significant difference between (i) the monolayer culture (2D) and the spheroids (3D) (+++ *p*< 0.001) and (ii) the first day (at day 3) of measurement and all subsequent days (* *p* < 0.05, ** *p* < 0.01, *** *p* < 0.001); was calculated in GraphPad Prism 6, by the two-way ANOVA considering Dunnett’s multiple comparisons test. Results are presented as the mean ± SD (N = 3). The dotted line denotes the expression of the corresponding gene in monolayer culture (1-fold change); the grey line indicates up- or downregulation of genes with the threshold set at 1.5-fold, which is more than 1.5 or less than 0.66 relative expression, respectively.

**Figure 5 cells-09-02557-f005:**
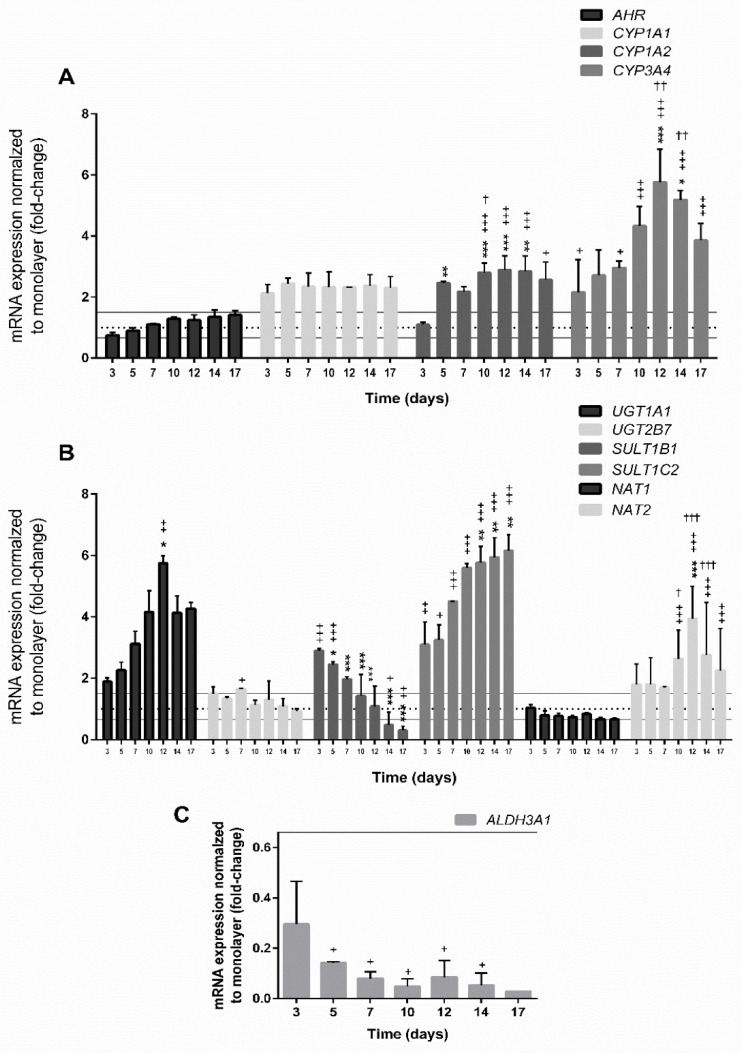
The mRNA expression of selected genes involved in the I (**A**) and II (**B**,**C**) phase of metabolism over time from day 3 to day 17. The significant difference between (i) the monolayer culture (2D) and the spheroids (3D) (+ *p* < 0.05, ++ *p* < 0.01, +++ *p*< 0.001); (ii) the first day (at day 3) of measurement and all subsequent days (* *p* < 0.05, ** *p* < 0.01, *** *p* < 0.001); and (iii) within days († *p* < 0.05, †† *p* < 0.01, ††† *p* < 0.001) was calculated in GraphPad Prism 6, by two-way ANOVA considering Dunnett’s multiple comparisons test. Results are presented as the mean ± SD (N = 3). The dotted line denotes the expression of the corresponding gene in monolayer culture (1-fold change), the grey line indicates up- or downregulation of genes with the threshold set at 1.5-fold, which is more than 1.5 or less than 0.66 relative expression, respectively.

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
