# Peer review of "Characterization of In Vitro 3D Cell Model Developed from Human Hepatocellular Carcinoma (HepG2) Cell Line"

_cells, 2020, doi:10.3390/cells9122557_

Round 1

Reviewer 1 Report

This review is a comprehensive description of a HepG2 spheroid 3D model and provides a detailed comparison to commonly used monolayer models that unfortunately are more distant from in vivo conditions hepatocytes present compared to the proposed 3D model. 

This is a clearly written paper with adequate description of the methods and coherent results. This paper can be used as basis for further studies using 3000c/spheroid 3D HepG2 models.

Although overall well written, there are some minor language deficits that should be addressed. Following are a few examples from the text. Starting from line 32 "In 3D models are better accomplished the characteristics mimicking human tissues, in particular liver, which is the main target organ of toxification and detoxification processes [1,2]" the sentence is a bit difficult to understand due to its structure. The suggestion would be to edit it to "3D models better reflect the characteristics mimicking human tissues, in particular liver, which is the main target organ of toxification and detoxification processes". Line 223 " In opposite to this, the spheroids with an initial density..." it would be better to write "in contrast" or "oppositely".

Best regards!

Author Response

Dear Reviewer,                                                              23rd of November, 2020

We would like to thank you for your valuable suggestions and for the opportunity to revise the manuscript (cells-993580) entitled “Characterization of in vitro 3D cell model developed from human hepatocellular carcinoma (HepG2) cell line”. We have carefully considered all the comments and below you will find our response to each comment. All the relevant data suggested by the reviewers were added and/or changed in the paper and are highlighted in track changes.

On the behalf of the authors,                                                                              

Yours sincerely,

Dr. Bojana Žegura  

#rev1

  1. This review is a comprehensive description of a HepG2 spheroid 3D model and provides a detailed comparison to commonly used monolayer models that unfortunately are more distant from in vivo conditions hepatocytes present compared to the proposed 3D model. This is a clearly written paper with adequate description of the methods and coherent results. This paper can be used as basis for further studies using 3000c/spheroid 3D HepG2 models.

A: We thank the reviewer for a very positive response.

  1. Although overall well written, there are some minor language deficits that should be addressed. Following are a few examples from the text. Starting from line 32 "In 3D models are better accomplished the characteristics mimicking human tissues, in particular liver, which is the main target organ of toxification and detoxification processes [1,2]" the sentence is a bit difficult to understand due to its structure. The suggestion would be to edit it to "3D models better reflect the characteristics mimicking human tissues, in particular liver, which is the main target organ of toxification and detoxification processes". Line 223 "In opposite to this, the spheroids with an initial density..." it would be better to write "in contrast" or "oppositely".

Best regards!

A: Thank you for your suggestions. We rewrote the sentence in line 32 ("The 3D cell models better reflect the characteristics mimicking human tissues, in particular liver, which is the main target organ of chemical activation and detoxification processes") and changed the word "In opposite" to "In contrast" (line 140). We also changed the subtitle from "2.5. Quantification and viability determination of the whole spheroid by Live/Dead staining" to "2.5. Analyses of cell cycle and cell proliferation by flow cytometry".

Reviewer 2 Report

The study presents the characterisation of HepG2 cell line grown as spheroids as a potential 3D platform for genotoxicity test. The work is well introduced and described, with solid methods and correct references. This is an important topic that needs optimisation of reliable and relevant 3D models for in vitro testing. Nevertheless, the study presented here shows a basic characterisation of a 3D culture technique that has been used for several years and that the Authors did not apply to a specific scientific hypothesis. The vast majority of the data presented has been already described in other publications, as clearly stated in the Manuscript at several points in the Results section. This shows the limited novelty of the study. Most of these results were already known or predictable based on the literature. The parameters for the characterisation were not contextualised. To determine if a 3D model is suitable for genotoxicity testing the authors should define what are the ideal characteristics of a 3D model. The Authors did not explain why the chosen time points and relative phenotype would be relevant for genotoxicity assessment. This should have been shown in actual genotoxicity tests with chemicals in vitro.

For these reasons, I believe that the manuscript should not be published on Cells.

I enclose some of the major comments.

  • In the introduction, what does ‘toxification’ mean in the context of the liver?
  • In the introduction, the authors claim that “The most important advantages of culturing cells in the form of 3D are increased cell-cell and cell-matrix interactions”. This is not true for spheroids that don’t contain matrix, unless the Authors describe the difference among 3D cultures or determine if HepG2 spheroids produce their own matrix.
  • Introduction: “Although 3D cell models are superior to 2D cell models, the lack of standardization of 3D cell culture protocols and insufficient characterisation of the spheroids prevents the integration of 3D cell culture models into the field of toxicology research particularly due to variability in structure and composition of the formed spheroids [38,39].” Characterisation and standardisation of HepG2 spheroid culture has been described in other papers referenced in this study. If the Authors refer to specific cell features that need to be further characterised, this should be better explained.
  • The results on growth and morphological changes over time are not novel. Other studies report similar analysis, i.e. in [18,32]. If there are significant differences, this should be emphasised in the text.
  • Does the surface area of spheroids keep increasing/changing after 7 days of culture?
  • Figure 1 should be a supplementary because it is a basic characterisation of spheroid size.
  • In the Live/Dead analysis, dead cells after 10-12 days in 3000 cells/spheroid condition are not in the centre, but mainly in the outer layers. What is the meaning of this distribution? The Authors should not claim that these cells are in the centre, or should show a different representative image. The Authors also claim that “There are only a few studies so far that specifically determined the size or time at which hepatic spheroids develop necrosis that is associated with the cell type, cell number, and culture conditions [19,32,52].” Are there differences between the results shown in this study and previously published results? Without discussing the results with the literature, it is really hard to understand the relevance or novelty of this characterisation.
  • Figure 2 and figure 3 should be combined into a single figure since they relate to the same test.
  • The sentence “Previous studies on 3D models reported a strong decrease in cell proliferation, which is associated with a time-dependent cell differentiation process [32,53,54].” What 3D models are we talking about here? With what cell lines? Is this fact positive or negative for the aim of the model?
  • The sentence “Similarly, a decrease in the number of proliferating cells in HepG2 spheroids and spheroids cultured in hydrogels was reported by Ramaiahgari et al. (2014) and Lee et al. (2009).” shows that similar results and culture conditions were already performed. This is mentioned several times about the results, mining the novelty of the study and questioning the relevance of the work.
  • Figure 4C is not mentioned in the text.
  • It is not clear what is the relevance of the section on cell cycle (Figures 4C and 4D).
  • Why was gene expression performed on 2D cultured HepG2 for 2 days only instead of longer time points? These cells can easily grow in 2D for several passages and days. It is not true that “HepG2 can not be cultured long term due to confluence that limits the duration of culturing. “
  • What is the relevance of a slight increase in CDKN1A expression in 3D after 3 days if the overall expression is significantly lower than 2D cultures?
  • The Authors claim that “Live/Dead stained spheroids (density 3.000 cells/spheroid) showed slight signs of necrosis in the centre, which could be related to the hypoxia as reported by Ramaiahgari et al. (2014) (Figure 5).” This is the opposite of what the Authors describe in Figure 2-results where it is stated that there is no necrotic core at this cell density.
  • Different results in the Gene Expression section are described as ‘increased’ ‘decreased’ or ‘stable’ but are not supported by statistics. If the difference is not significant (in respect to the correct control) it should not be described as an actual change.

Author Response

Dear Reviewer,                                                           23rd of November, 2020

We would like to thank you for your valuable suggestions and for the opportunity to revise the manuscript (cells-993580) entitled “Characterization of in vitro 3D cell model developed from human hepatocellular carcinoma (HepG2) cell line”. We have carefully considered all the comments and below you will find our response to each comment. All the relevant data suggested by the reviewers were added and/or changed in the paper and are highlighted in track changes.

On the behalf of the authors,                                                                              

Yours sincerely,

Dr. Bojana Žegura  

#rev2

  1. The study presents the characterisation of HepG2 cell line grown as spheroids as a potential 3D platform for genotoxicity test. The work is well introduced and described, with solid methods and correct references. This is an important topic that needs optimisation of reliable and relevant 3D models for in vitro testing. Nevertheless, the study presented here shows a basic characterisation of a 3D culture technique that has been used for several years and that the Authors did not apply to a specific scientific hypothesis. The vast majority of the data presented has been already described in other publications, as clearly stated in the Manuscript at several points in the Results section. This shows the limited novelty of the study. Most of these results were already known or predictable based on the literature. The parameters for the characterisation were not contextualised. To determine if a 3D model is suitable for genotoxicity testing the authors should define what are the ideal characteristics of a 3D model. The Authors did not explain why the chosen time points and relative phenotype would be relevant for genotoxicity assessment. This should have been shown in actual genotoxicity tests with chemicals in vitro.

For these reasons, I believe that the manuscript should not be published on Cells.

I enclose some of the major comments.

 A: We thank the reviewer for the response. Below we will provide comprehensive answers to the comments.

  1. In the introduction, what does ‘toxification’ mean in the context of the liver?

A: Thank you for the remark. We replaced "toxification" with "chemical activation" (line 33) to be clearer.

  1. In the introduction, the authors claim that “The most important advantages of culturing cells in the form of 3D are increased cell-cell and cell-matrix interactions”. This is not true for spheroids that don’t contain matrix, unless the Authors describe the difference among 3D cultures or determine if HepG2 spheroids produce their own matrix.

A: It is commonly known that cells in 3D conformation have enhanced cell–cell interactions, cell–ECM interactions and thus can produce their own matrix, which was shown in the literature that we cited in our manuscript (Li et al., 2008; Aucamp et al., 2018; Ramaiahgari et al., 2014; Elje et al., 2019; Loessner et al. 2010; Wrzesinski et al., 2015; Štampar et al., 2019). In addition, in our study, we used 4 % methylcellulose as a matrix to promote spheroid generation and prevent the formation of cell monolayers in a highly reproducible manner. However, the methylcellulose is degraded at 37 °C, and thus consequently after degradation the HepG2 cells in 3D conformation start to produce their own matrix as shown in the literature.

  1. Introduction: “Although 3D cell models are superior to 2D cell models, the lack of standardization of 3D cell culture protocols and insufficient characterisation of the spheroids prevents the integration of 3D cell culture models into the field of toxicology research particularly due to variability in structure and composition of the formed spheroids [38,39].” Characterisation and standardisation of HepG2 spheroid culture has been described in other papers referenced in this study. If the Authors refer to specific cell features that need to be further characterized, this should be better explained.

A: There were some attempts to characterize hepatic 3D models (e.g. HepG2, HepaRG), however, this was done only partly, mostly showing morphological changes, changes in the surface area and thickness of spheroids, fluorescent imaging of the spheroids and immunostaining for albumin expression. In our study, we showed the changes in morphology and in the growth of spheroids. We performed confocal microscopy for the determination of live/dead cells and also quantified the fluorescent images (Z-stacks). Moreover, we showed time dependent changes in the cell cycle and decrease in cell proliferation, which was not done before in the respective model and with the methodologies applied in our study. For this purpose, we applied a novel approach using a flow cytometry, where two endpoints (cell cycle distribution and proliferation marker KI67) were simultaneously determined in the same cell population, which was never done before on spheroids. In addition, for the first time the expressions of genes involved in cell proliferation, a wide specter of genes involved in phase I and II of the metabolism and genes encoding hepatic markers were shown. Therefore, we believe that the study represents a great deal novelty in the field of 3D cell models.

  1. The results on growth and morphological changes over time are not novel. Other studies report similar analysis, i.e. in [18,32]. If there are significant differences, this should be emphasised in the text.

A: We agree with the reviewer, that growth and morphological changes of 3D spheroids were already presented in the literature, however, we had to show what is happening when culturing the HepG2 spheroids (this particular model) with the initial density of 3.000 and 6.000 cells/spheroid to confirm that our study is consistent with already published data. This enabled us to decide, which initial density is the most appropriate for further experiments and characterization (cell cycle distribution measurements, proliferation and gene expressions). In the manuscript (page 6, line 239) we also stated which initial density is more appropriate:  “Based on these results, we concluded that 3.000 cells per well was a more optimal density for the formation of spheroids that could be used for long-term exposures; therefore, this density was selected for further characterisation of the spheroids.” and (page 8, line 325): “Our results showed that spheroids with a lower initial density maintain the proliferation of cells at a higher rate and over a longer period compared to the spheroids with a higher initial density, meaning that spheroids with a lower initial density are more suitable for long-term exposures.” In line 343 we also added: “These results again show that spheroids with lower initial density maintain higher rate of cell proliferation compared to spheroids with higher initial density.”

  1. Does the surface area of spheroids keep increasing/changing after 7 days of culture? Figure 1 should be a supplementary because it is a basic characterization of spheroid size.

A: Unfortunately, we measured the growth of spheroids (surface area) only until day 7. We agree that the measurements represented a basic characterization of spheroid size, although it is important for the understanding how the cells in the 3D model behave, therefore, we concur with the suggestion to move the Figure 1 to the supplementary. In the manuscript we accordingly changed the name of the “Figure 1” to “Figure S1” and in page 5 line 225 we changed “Figure 1A-B” to “Figure 1S A-B”, in line 226 “Figure 1C-D” to “Figure 1S C-D”, in line 228 we deleted “Figure 1”, in line 229 “Figure 1A” to “Figure 1S A”, in line 231 “Figure 1B” to “Figure 1S B”, in line 232 “Figure 1C” to “Figure 1S C” and in line 234 “Figure 1D” to “Figure 1S D”.

  1. In the Live/Dead analysis, dead cells after 10-12 days in 3000 cells/spheroid condition are not in the centre, but mainly in the outer layers. What is the meaning of this distribution? The Authors should not claim that these cells are in the centre, or should show a different representative image. The Authors also claim that “There are only a few studies so far that specifically determined the size or time at which hepatic spheroids develop necrosis that is associated with the cell type, cell number, and culture conditions [19,32,52].” Are there differences between the results shown in this study and previously published results? Without discussing the results with the literature, it is really hard to understand the relevance or novelty of this characterisation.

A: For the microscopic analysis we used confocal laser scanning microscope and we took z-Stacks of optical sections through the entire volume of the spheroids using appropriate excitation and emission settings for simultaneous dual-channel recordings of FDA and PI. The Z-stacks were presented as a ‘maximum intensity projection image’ gallery. A maximum intensity projection is a scientific visualization technique that takes 3D data (in our case a Z-stack of confocal microscope images) and turns it into a single 2D image. The projection takes the brightest pixel (voxel) in each layer and displays that pixel intensity value in the final 2D image. Commonly the projection is created so that it is being viewed “down” into the image stack along the Z-axis. Therefore, the dead cells are occurring inside the spheroid. In case of 3.000 cells/spheroid there is no hypoxia detected until day 12 (very little cells in the center) and the cells stained in red are random death cells as cell death is the naturally occurring process. In the case of 3.000 cells/spheroid we do not relate these cells with the necrotic core. We also corrected in the Discussion part the explanation that in the case of 3.000 cells/spheroid the necrotic core occurs. This was actually the case for 6.000 cells /spheroid.  Based on the obtained data from 6.000 cells/spheroid we could assume that dead cells will appear in the center and the amount of death cells will increase with time. When we use 3D cell model for genotoxicity testing, we do not want to have a necrotic core particularly after long time of cultivation (10-12 days), as in this way we can use the model for studying prolonged chronic exposures, which is more relevant for real life human exposures.  In case 3D cell models are used for studies related to cancer, then the necrotic core actually better mimics the real situation occurring in tumors. This we actually explained in the manuscript (line 263- 267). Therefore, in our study, for better understanding what is happening through the entire spheroid, we conducted the advanced quantification of the single stacks to get more representative results, which can contribute to new knowledge in this field. And again to mention this was not done before in HepG2 3D cell model.

As cited in the manuscript there are three studies that briefly mentioned the formation of necrotic core in spheroids. Now we added more information on these studies and discussed them in term of the results obtained in our study results. The added text (page 6, line 271) is as follows: “…; therefore the present study contributes new knowledge on the formation of necroitic core in spheroids with time of cultivation. Previously, in encapsulated 3D HepG2 aggregates no necrotic core was observed up to three days of culturing, while with prolonged cultivation the thicknes of aggregates disabled determination of necrotic cells in the center of 3D cell model [19,32,52] reported a small necrotic core in HepG2 3D spheroids developed in hanging drops after 1 day of culturing, which was more represented as separate dead cells; however, the viability was stable over time and the cells were cultured for up to 21 days. Further, in hepatic C3A spheroids with initial density of 2500 cells/spheroid small patches of cell death started to occur approximately at day 14 and, by day 18, a necrotic core was formed [52]. Altogether these studies including ours show that necrosis occurs in the core of the spheroid with time of cultivation and depends on the cell type and 3D conformation.”

  1. The Authors also claim that “There are only a few studies published so far that specifically determined the size and time of spheroid growth (52, 19, 32) however until now, non of the studies showed at which culture time and initial cell density hepatic spheroids develop necrosis.

A: Yes, this is true. There is no clear information when exactly the necrosis occurs. It is also important to mention that this depends on the cell type, initial density of cells for the formation of spheroids (in more dense spheroids necrosis occur fasted that if we have lower initial density. The present study clearly showed that at lower density (3000 cells/ spheroid) there was no necrosis in the center of the spheroid detected at day 7, while at a higher density (6000 cells/spheroid) we can see the appearance of a necrotic center after day 6. However, thank you for the remark, we again conducted the statistical analysis with the unpaired t- test comparing all days to the first day of measurement (see below) and we added new calculated significant marks (*) to the graphs.

Low density: [3-5; Unpaired t test, P value 0,9089, P value summary ns, Significantly different (P < 0.05)?No, One- or two-tailed P value? Two-tailed t, df t=0,1218, df=4]. [3-7; Unpaired t test, P value 0,0637, P value summary ns, Significantly different (P < 0.05)? No,  One- or two-tailed P value? Two-tailed, t, df t=2,544, df=4]. [3-10; Unpaired t test, P value 0,0003, P value summary***, Significantly different (P < 0.05)? Yes, One- or two-tailed P value? Two-tailed t, df t=11,34, df=4]. [3-12; Unpaired t test, P value 0,0382, P value summary*, Significantly different (P < 0.05)? Yes, One- or two-tailed P value? Two-tailed t, df t=3,548, df=3].

and higher density: [3-6; Unpaired t test, P value 0,0472, P value summary*, Significantly different (P < 0.05)? Yes, One- or two-tailed P value? Two-tailed t, df t=2,832, df=4]

In addition, we again reviewed the literature and have now added discussion on published data and necrosis. We added the paragraph on discussion on the formation of necrotic core in published literature and is now as follows (Page 6, line 269): “There are only a few studies so far that specifically determined the size or time at which hepatic spheroids develop necrosis that is associated with the cell type, cell number, and culture conditions [32,52]; therefore the present study contributes new knowledge on the formation of necrotic core in spheroids with time of cultivation [19,32,52]. Previously, in encapsulated 3D HepG2 aggregates no necrotic core was observed up to three days of culturing, while with prolonged cultivation the thickness of aggregates disabled determination of necrotic cells in the center of 3D cell model [19,32,52] reported a small necrotic core in HepG2 3D spheroids developed in hanging drops after 1 day of culturing, which was more represented as separate dead cells; however, the viability was stable over time and the cells were cultured for up to 21 days. Further, in hepatic C3A spheroids with initial density of 2500 cells/spheroid small patches of cell death started to occur approximately at day 14 and, by day 18, a necrotic core was formed [52]. Altogether these studies including ours show that necrosis occurs in the core of the spheroid with time of cultivation and depends on the cell type and 3D conformation.”

  1. Figure 2 and figure 3 should be combined into a single figure since they relate to the same test.

A: Thank you for the suggestion. We combined Figure 2 and 3 and renamed the new figure into Figure 1. Accordingly we re-numbered all other Figures.

  1. The sentence “Previous studies on 3D models reported a strong decrease in cell proliferation, which is associated with a time-dependent cell differentiation process [32,53,54].” What 3D models are we talking about here? With what cell lines? Is this fact positive or negative for the aim of the model?

A: Thank you for the remark. To be clearer, we added »HepG2« in the sentence (Page 8 line 304) and deleted one reference (55). Now the sentence is: “Previous studies on HepG2 3D models reported a strong decrease in cell proliferation, which is associated with a time-dependent cell differentiation process [32, 54].” Just to explain this statement. It is known that the proliferation rates of HepG2 cells cultured in 3D and 2D are different (Edmondson et al. 2014), showing reduced proliferation rates in 3D cultures compared to those cultured in 2D, see literature (Elje et al., 2019, Luckert et al., 2017). Moreover, it was also shown that many other human cell lines (CaCo-2 (Luca et al. 2013), DLD-1 (Luca et al. 2013), HT-29 (Luca et al. 2013), HEK 293 (Wong et al., 2007), MCF10A (Wang et al., 2010)) have decreased proliferation or proliferate slowler in 3D formation compared to 2D. According to this, the reduced proliferation rate of cells better represents the growth of cells in vivo compared to those cultured in an unnatural 2D environment.

  1. The sentence “Similarly, a decrease in the number of proliferating cells in HepG2 spheroids and spheroids cultured in hydrogels was reported by Ramaiahgari et al. (2014) and Lee et al. (2009).” shows that similar results and culture conditions were already performed. This is mentioned several times about the results, mining the novelty of the study and questioning the relevance of the work.

A: As described, the proliferation was performed at particular culturing times and with different initial cell densities to comprehensively validate and characterize the particular model, which is very important before such models can be routinely used for genotoxicity assessment. To claim without experimental confirmation that our model behaves in the same way as described in the literature for some other 3D cell models would not be scientifically correct. With this statement, we wanted to show that the obtained results on the newly developed HepG2 3D model are in line with those already published, which confirms the accuracy of our model.

  1. Figure 4C is not mentioned in the text.

A: Thank you for the remark, we added Figure 4C (now 2C, due to figure changes) in the text page 9 line 331 »Figure 2C«.

  1. It is not clear what is the relevance of the section on cell cycle (Figures 4C and 4D).

A: Cell cycle is very much related to the cell proliferation. It is commonly known that cell cycle of proliferating eukaryotic cells consists of four phases, namely G1, S, G2, and M. The cells that do not proliferate are arrested in G0 phase. Thus, in our study we clearly showed that with prolonged exposure when HepG2 cells in 3D stopped to proliferate (shown by determination of proliferation marker Ki67 in cell population) they accumulated in G0 phase. This analysis was enabled by using a novel approach where we stained the same cell population with anti-Ki67 antibodies for determination of cell proliferation and Hoechst 33258 for cell cycle analysis, which was not done before (again showing novelty of our study). Thus, this enables to follow the same cell for two different end-points. The Figure 2D shows the overlay of two different end-points measured simultaneously, namely staining with Hoechst for the cell cycle (red) and anti-KI67 antibody (blue) for those cells proliferating in G0/G1 cells. Since the cells in G0 phase do not proliferate and are not Ki67 positive (are not marked in blue), the results clearly show that with time more cells are arrested in G0 phase.

  1. Why was gene expression performed on 2D cultured HepG2 for 2 days only instead of longer time points? These cells can easily grow in 2D for several passages and days. It is not true that “HepG2 can not be cultured long term due to confluence that limits the duration of culturing.“

A: Two days of culturing were selected, because usually in monolayer cultures various endpoints are measured after 48 hours (or 2 days) (e.g. Cells are first cultured for 24 hours to attach to the plastic and then exposed for 24 hours, which altogether takes 48 hours). In order to be this clear we included this information in the Section 2.6 in bracket and is as follows (Page 4, line 167): “(this is usually the time when in 2D the gene expression is evaluated after the exposure to various compounds; 24 hours for cells to attach to the surface of the plastic and 24 hours of subsequent exposure)”. Here we also added missing information on the isolation of mRNA from monolayer culture line 170: “…one T25 plate in case of 2D and from..” and “… in case of 3D…” .

We agree with the reviewer that HepG2 cells can be grown for a longer period of time in 2D conditions, however, after growing for more than a week on the flat surface, the cells are no longer in monolayer culture, but they grow one over the other and form 3D shapes and thus acquire new properties, which was already described in the literature by Luckert et al., 2017. However, as the round shape is the most optimal in terms of surface area to volume and due to the cell-to-cell adhesion, it is the most appropriate for the cells that grow in culture for longer period of time and form 3D structures to grow in spheroidal shape.

  1. What is the relevance of a slight increase in CDKN1A expression in 3D after 3 days if the overall expression is significantly lower than 2D cultures?

A: We agree that there is no significant difference, therefore, we changed the sentence (Page 10, line 387) to “We noticed a slight increase in the CDKN1A expression with time of incubation reaching the maximal level at day 7 (-7.14-fold) (Figure 3B).”

  1. The Authors claim that “Live/Dead stained spheroids (density 3.000 cells/spheroid) showed slight signs of necrosis in the centre, which could be related to the hypoxia as reported by Ramaiahgari et al. (2014) (Figure 5).” This is the opposite of what the Authors describe in Figure 2-results where it is stated that

A: Thank you for the remark, we agree that there is no necrotic core at this cell density. We changed the sentence to »Live/Dead stained spheroids (density 3000 cells/spheroid) showed no signs of necrosis in the center, however, in spheroids with a higher initial density (6000 cells/spheroid) the necrosis was observed at day 6. This could be related to the hypoxia as reported by Ramaiahgari et al. (2014) (Figure 3).”

  1. Different results in the Gene Expression section are described as ‘increased’ ‘decreased’ or ‘stable’ but are not supported by statistics. If the difference is not significant (in respect to the correct control) it should not be described as an actual change.

A: When considering positive result it has to be pointed out that the statistical significance is not the determining factor for positive result, but is 1.5-fold up/down-regulation (relative expression >1.5 or <0.66, respectively) as already stated in the Section 2.6. However, from the Figures 4-5 it is clear that statistical analysis was performed and it can be seen which samples were statistically different from control group. In the text we now added the information if samples were significantly different from control group or not.

Reviewer 3 Report

Characterization of in vitro 3D cell model developed 2 from human hepatocellular carcinoma (HepG2) cell line

The manuscript gives a comprehensive characterization of a 3D liver cell model. The work appears to be of good quality and is reasonably well written. Development of more advanced in vitro model is important in order to establish methodology that can replace use of animals and are more organ like. I have a few points that should be addressed by the authors.

There are published a few articles about characterization of HepG2 spheroid model, and you should more clearly address the novelty of your work and how your work build upon previous works. A weakness in your work is the lack of protein expression data, which could have confirmed that the increased gene expressions are revealed at higher cellular levels. This should be addressed in the text.

The authors emphasize the advantage of using 3D models compared to 2D models, and it is claimed that 2D-models may give false results. This may of course also apply to a 3D-model, it all depends on the purpose of the experiments. For example, it may be useful to test cells with extensive proliferation as with low confluence 2D-mononlayer or 2D-models for screening purposes. It should be addressed that any cell model should be used with some cautions and that there is both pros and cons with the different models.

In the method and result section you compare your gene expression pattern of the 3D model with the 2D model. I do not understand how you normalize the 3D result with the 2D result. E.g. is gene expression in 3D model after 10 days normalized with a 2D monolayer cultivated for 10 days. Is it based on mRNA extraction from a certain number of cells or mRNA extracted from a certain concentration of DNA? I am worried that your results can give a false impression of an increased gene expression due to a comparison between apple and pears. This must be explained better in the text.

In the text you have a statement that confluency in 2D-models limit the usefulness of long-term exposure. Can same argument be used on 3D-models? It appears that the viability of cells in spheroids is reduced as a function of time, and you also observe reduced cell proliferation. Same is observed with confluent 2D models.

Author Response

Dear Reviewer,                                                           23rd of November, 2020

We would like to thank you for your valuable suggestions and for the opportunity to revise the manuscript (cells-993580) entitled “Characterization of in vitro 3D cell model developed from human hepatocellular carcinoma (HepG2) cell line”. We have carefully considered all the comments and below you will find our response to each comment. All the relevant data suggested by the reviewers were added and/or changed in the paper and are highlighted in track changes.

On the behalf of the authors,                                                                              

Yours sincerely,

Dr. Bojana Žegura  

#rev 3

Comments and Suggestions for Authors

Characterization of in vitro 3D cell model developed from human hepatocellular carcinoma (HepG2) cell line

The manuscript gives a comprehensive characterization of a 3D liver cell model. The work appears to be of good quality and is reasonably well written. Development of more advanced in vitro model is important in order to establish methodology that can replace use of animals and are more organ like. I have a few points that should be addressed by the authors.

 A: We thank the reviewer for a very positive response.

  1. There are published a few articles about characterization of HepG2 spheroid model, and you should more clearly address the novelty of your work and how your work build upon previous works. A weakness in your work is the lack of protein expression data, which could have confirmed that the increased gene expressions are revealed at higher cellular levels. This should be addressed in the text.

 A: Thank you for the suggestions about the novelty, we changed some sentences in the manuscript to highlight the importance and novelty of the study. In section results and discussion, we added a sentence: page 5 line 211 “ However, before 3D cell models can be integrated for genotoxicity testing research, there is a need for development and subsequent standardization of robust models that accurately predict the possible effects of studied compounds [45].”, page 5 line  215 “To our knowledge the present study is the first where a comprehensive characterization of the HepG2 3D cell model was performed.” and in conclusions we added a sentence in page 15 line 571: “To our knowledge, the present study is the first where 3D HepG2 cell model was systematically characterized and standardized including advanced cell cycle and proliferation analysis by flow cytometry and gene expressions.” And in line 585: “…particular for sulfotransferases in phase II, thus indicating differentiation into more metabolically competent cells, which however has to be further confirmed at the protein level.” As we used in the study several approaches (Z-stack imaging and quantification of Z-stacks), simultaneous staining of cells in spheroids for cell proliferation (Ki67 marker) and cell cycle (Hoechst 33258 dye), comprehensive analysis of the expression of several genes at different ages of the spheroids which added greatly to the knowledge on the expression on these genes, we believe that the study shows a high level of novelty.

We agree that the measurements of proteins would give a better idea of what is happening with the protein expressions. Unfortunately, we performed only the measurements of gene expression and we can state only what is happening at the transcriptomic levels.  As the expression of proteins actually reflects what is happening in the cell, we will consider the measurement of proteins in the following studies.  In the conclusion section (Lines 583-587) we added the importance of protein analysis: “ Moreover, the spheroids revealed increased liver-specific functions and demonstrated strong physiological relevance concerning gene expression of hepatic markers and metabolic enzymes, in particular for sulfotransferases in phase II, thus indicating differentiation into more metabolically competent cells, which however has to be confirmed at the protein level.”

  1. The authors emphasize the advantage of using 3D models compared to 2D models, and it is claimed that 2D-models may give false results. This may of course also apply to a 3D-model, it all depends on the purpose of the experiments. For example, it may be useful to test cells with extensive proliferation as with low confluence 2D-mononlayer or 2D-models for screening purposes. It should be addressed that any cell model should be used with some cautions and that there is both pros and cons with the different models.

A: Thank you for the remark. In conclusions we added the sentence in page 15 line 549: “Nevertheless, before 3D models can be routinely used for genotoxicity assessment they have to be comprehensively characterized and growth conditions need to be optimized to allow the reproducibility and comparability of the results. Furthermore, systematic characterisation allows us to identify all the crucial advantages and disadvantages, which is of high importance for the further use of the model.”

  1. In the method and result section you compare your gene expression pattern of the 3D model with the 2D model. I do not understand how you normalize the 3D result with the 2D result. E.g. is gene expression in 3D model after 10 days normalized with a 2D monolayer cultivated for 10 days. Is it based on mRNA extraction from a certain number of cells or mRNA extracted from a certain concentration of DNA? I am worried that your results can give a false impression of an increased gene expression due to a comparison between apple and pears. This must be explained better in the text.

A: As already described in the comments to the reviewer 2, two days of culturing were selected, because usually in monolayer cultures various endpoints are measured after 48 hours (or 2 days) (e.g. cells are first cultured for 24 hours to attach to the plastic and then exposed for 24 hours, which altogether takes 48 hours). Therefore, in the study we took this time point as a starting point for the comparison of gene expression in spheroids that were cultured for ≥ 3 days. We think that we are not comparing the apples and pears, as the cell line that was used for all experiments is the same in 2D and 3D – so HepG2 cells. Only the time of culturing is different, where for 2D we cultured the same cell line (for 2 days) as for 3D (cultured for ≥ 3 days). Thus we compared the relative amount of mRNA in 2D to that in 3D of the same cell line.

Also the fact that cells grown in 2D monolayer for 10 days are not a monolayer culture any more but start to grow in 3D, which was already discussed by Luckert et al., 2017. However, as the round shape is the most optimal in terms of surface area to volume and due to the cell-to-cell adhesion, it is the most appropriate for the cells that grow in culture for longer period of time and form 3D structures to grow in spheroidal shape. In the Section 3.4. (Line 353) we corrected the text to be more clear: “Data for the individual gene were compared to the expression of genes from the monolayer culture at the age of 2 days and are presented as the ratio between 2D (at 2 days of culturing) and 3D at the corresponding time point.” The same approach was already used in several previously published studies (Chang & Hughes-Fulford, 2009), (Ramaiahgari et al., 2014), (Shah et al., 2018).

In all experiments, 1 mg of total mRNA for each sample was used for reverse transcription to cDNA as stated in the Section 2.6. This is a common procedure in quantitative PCR. In the Section 2.6, we also state that mRNA was isolated from the pool of 25 spheroids, while we now added the missing information for the monolayer culture, which was one T25 plate. We also added in the text (Page 13, line 528: “In the present study, many crucial genes that are involved in the activation and detoxification of xenobiotic substances and are in HepG2 monolayer cultures expressed at a very low rate or are even not detectable, were clearly expressed in HepG2 3D spheroids.”

  1. In the text you have a statement that confluency in 2D-models limit the usefulness of long-term exposure. Can same argument be used on 3D-models? It appears that the viability of cells in spheroids is reduced as a function of time, and you also observe reduced cell proliferation. Same is observed with confluent 2D models.

A: Over time, proliferation does decrease in 3D conformation, but this more accurately reflects physiologically relevant situation to in vivo conditions as in liver, cells do not proliferate all the time. We showed that in spheroids with initial density of 3000 cells/spheroid no necrotic core and no increased amount of dead cells were noticed up to 17 days in culture. Thus in HepG2 3D cell model (with initial cell density of 3000 cells) there was no drop of viability of cells. In the text there was a mistake and we corrected the statement. It is now as follows (Page 11, line: 411): “Live/Dead stained spheroids (density 3000 cells/spheroid) showed no signs of necrosis in the centre, however, in spheroids with a higher initial density (6000 cells/spheroid) the necrosis was observed at day 6.

In opposite, in monolayer cultures, it is not possible to culture HepG2 cells for more than a week without the changes in cell morphology, detachment of the cells from the surface and most importantly increased drop in cell viability. In addition to that, it is now already commonly accepted that in 2D monolayer cultures the cells lack numerous biological functions like cell–cell and cell–matrix contacts, which results in decreased cell differentiation, flattened morphology of cells with altered cytoskeleton, reduced viability, and altered cell signaling pathways and most importantly reduction or loss of many hepatic enzymes involved in metabolism of xenobiotic substances, which was described in several papers such as Edmondson et al. 2014; Wrzesinski and Fey 2015, including ours Štampar et al., 2019 and Štampar et al., 2020 to give just a few examples. Moreover, it is increasingly recognized that cells grown in 3D environment more closely represent normal cellular functions due to the improved cell-to-cell and cell-to-matrix interactions, and by mimicking the in vivo architecture of natural tissues and organs (just a few examples of publications Zhang and Yang 2011; Fey and Wrzesinski 2012; Gunness et al. 2013 etc). Moreover, 3D cell cultures can be grown undisturbed over a longer period of time compared to monolayer cultures, which makes them an appropriate model for long-term repeated dose studies (Wong et al. 2011; Štampar et al., 2020). Thus there is no doubt that 3D cell models are gaining on importance as they more accurately represent in vivo conditions than 2D cell models.

Round 2

Reviewer 2 Report

Congratulations to the Authors for the work done on the manuscript. 

They have addressed all my comments and in my opinion the study can now be accepted for publication.